# Computer-Aided Diagnosis Improves the Detection of Clinically Significant Prostate Cancer on Multiparametric-MRI: A Multi-Observer Performance Study Involving Inexperienced Readers

**DOI:** 10.3390/diagnostics11060973

**Published:** 2021-05-28

**Authors:** Valentina Giannini, Simone Mazzetti, Giovanni Cappello, Valeria Maria Doronzio, Lorenzo Vassallo, Filippo Russo, Alessandro Giacobbe, Giovanni Muto, Daniele Regge

**Affiliations:** 1Department of Surgical Sciences, University of Turin, 10126 Turin, Italy; 2Department of Radiology, Candiolo Cancer Institute, FPO-IRCCS, 10060 Candiolo, Italy; giovanni.cappello@ircc.it (G.C.); valeriadoronzio@hotmail.it (V.M.D.); lorenzovassallo1987@gmail.com (L.V.); filippo.russo@ircc.it (F.R.); daniele.regge@ircc.it (D.R.); 3Department of Urology, Humanitas Gradenigo, 10153 Turin, Italy; alessandrogiacobbe@yahoo.it; 4Department of Urology, Humanitas University, 10153 Turin, Italy; g.muto@tin.it

**Keywords:** computer aided diagnosis, prostate cancer, artificial intelligence, assisted reading

## Abstract

Recently, Computer Aided Diagnosis (CAD) systems have been proposed to help radiologists in detecting and characterizing Prostate Cancer (PCa). However, few studies evaluated the performances of these systems in a clinical setting, especially when used by non-experienced readers. The main aim of this study is to assess the diagnostic performance of non-experienced readers when reporting assisted by the likelihood map generated by a CAD system, and to compare the results with the unassisted interpretation. Three resident radiologists were asked to review multiparametric-MRI of patients with and without PCa, both unassisted and assisted by a CAD system. In both reading sessions, residents recorded all positive cases, and sensitivity, specificity, negative and positive predictive values were computed and compared. The dataset comprised 90 patients (45 with at least one clinically significant biopsy-confirmed PCa). Sensitivity significantly increased in the CAD assisted mode for patients with at least one clinically significant lesion (GS > 6) (68.7% vs. 78.1%, *p* = 0.018). Overall specificity was not statistically different between unassisted and assisted sessions (94.8% vs. 89.6, *p* = 0.072). The use of the CAD system significantly increases the per-patient sensitivity of inexperienced readers in the detection of clinically significant PCa, without negatively affecting specificity, while significantly reducing overall reporting time.

## 1. Introduction

Prostate Cancer (PCa) is the most frequently diagnosed cancer in men, accounting for about 26% of new cancer diagnoses [1]. Until publication of the 2020 European and American Urological Guidelines [2], the standard diagnostic procedure for PCa diagnosis included prostate specific antigen (PSA), digital rectal exam (DRE) and systematic transrectal ultrasound (TRUS) guided biopsy to confirm the presence of the tumor. This workup has shown several limitations, the most important being probably a high rate of over-diagnosis and over-treatment of clinically insignificant PCa [3].

Multiparametric-MRI (mpMRI) has become a key investigation tool for the diagnosis and management of prostate cancer (PCa) patients [4,5,6] MpMRI has both increased the detection of clinically significant disease, and reduced the number of unnecessary biopsies [7]. As an example, the PROMIS study reported that mpMRI, used as a triage test in biopsy-naïve patients, could allow 27% of them to avoid primary biopsy, thus reducing over-diagnosis of clinically insignificant PCa, while improving the detection of the clinically significant ones [8]. However, a systematic review by Futterer et al. [9] showed that mpMRI accuracy in detecting clinical significant disease varied considerably among different studies (44–87%), strongly depending on reader experience [10,11]. In the last decade, computer-aided diagnosis (CAD) systems have become an active area of research to help radiologists to report examinations, reduce reader variability and reading time [12,13,14,15,16,17]. Nelson et al. [14] showed that the average sensitivity for cancer detection of CAD systems was 86.8%, with a range from 47% to 98%. However, the output of a CAD system could be presented in different modalities (binary segmentations, probability maps, etc.), heavily affecting its performance when used by radiologists. A few studies evaluated the performance of CAD systems in a clinical setting, namely investigating their potential role in improving reader performance when reporting mpMRI for PCa detection.

A recent international, multi-reader study by Greer et al. [18] showed that probability maps provided by a CAD system can significantly improve sensitivity of readers to more subtle prostate lesions, though strongly reducing specificity. Hambrock et al. [19] demonstrated that accuracy in characterizing pre-annotated prostate lesions increases using the CAD system, especially for non-experienced readers. However, they developed a CAD system for lesion characterization and not for detection purposes. Giannini et al. [20] presented a reading modality in which experienced readers reported the likelihood map of the entire prostate generated by a CAD system, showing that sensitivity was significantly higher in the CAD-assisted reading than in the unassisted one when reporting lesions with Gleason score > 6 and/or diameter > 10 mm, reducing overall reporting times. It must be noted that for other consolidated applications [21,22], non-experienced radiologists more than senior radiologists benefit from CAD systems cancer. To our knowledge, there are no studies assessing the potential of a fully automatic CAD scheme for non-experienced readers in reporting PCa in a clinical setting, i.e., without having pre-annotated lesions. 

The main aim of this single institution multi-reader study is to assess the diagnostic performance of non-experienced readers, i.e., resident radiologists, when reporting assisted by the likelihood map of the prostate generated by a CAD system, and to compare the results with the unassisted interpretation. Secondary aims are the assessment of reading times and inter-observer variability between the two reading modalities.

## 2. Materials and Methods

### 2.1. Patients Population and Study Design

This retrospective study enrolled men that performed mp-MRI in one institution between November 2014 and November 2016. Inclusion criteria were: (a) age ≥50 years; (b) PSA ≥ 4 ng/mL; (c) mpMRI of the prostate; (d) either pathology-confirmed PCa or at least 2 years of follow-up in men with no evidence of PCa on MRI; (e) written informed consent. Patients with previous history of PCa were excluded. The study design was approved by the local Ethics Committee, in accordance with the Helsinki Declaration, and registered at ClinicalTrials.gov (identifier NCT04398173).

### 2.2. Multiparametric MRI

Images were acquired with a 1.5 T scanner (Optima MR450w, GE Healthcare, Milwaukee, Illinois, USA) using a 32-channel phased-array coil combined with air-inflated endorectal coil (Medrad, Indianola, Pa). Imaging included three orthogonal T2w sequences, axial Diffusion Weighted Imaging (DWI) and Dynamic Contrast Enhanced (DCE) which was triggered to start simultaneously with the power injection of 0.1 mmol/kg gadobutrol (Gadovist, Bayer Schering, Berlin, Germany) through a peripheral line at 0.7 mL/s, followed by infusion of 20 cc normal saline at the same rate. The average time to complete the whole MRI examination was 35 min. Acquisition parameters were detailed in Appendix A and satisfied the scanning requirements for prostate imaging [23].

### 2.3. Reference Standard

One experienced radiologist (>800 prostate mpMRI reported per year, 7 years of experience) reported all mpMRI examinations using the PIRADS v2.0 score [23]. In patients with PIRADS ≥ 3 and pathology-confirmed PCa, the radiologist mapped PCa findings to mpMRI, recording for each lesion its location, maximum diameter and PIRADS classification. Patients with PIRADS < 3 were considered negative at the reference standard if during the follow-up either one of the following conditions was met: (1) PSA doubling time was >3 years; (2) PSA doubling time was ≤3 years but follow-up mpMRI did not report any suspicious findings for PCa; or (3) PSA doubling time was ≤3 years and follow-up mpMRI triggered a prostate biopsy that was negative for PCa presence.

### 2.4. CAD System and Image Interpretation 

Three resident radiologists all with 1 year of experience in MRI (reader 1, 200 reports; reader 2, 120 reports; reader 3, 200 reports), blinded to disease prevalence in this cohort, reported all mpMRI examinations twice, more than 6 weeks apart to minimize recall bias.

First, readers were asked to report all mpMRI using the output of a previously described and validated CAD system [20,24]. In brief, it consists of a 3D color-coded map of the entire prostate, in which voxels are colored based on their likelihood of being malignant. This map is automatically generated by the CAD system using a support vector machine classifier fed with quantitative parameters derived from the mpMRI acquisition (i.e., the normalized T2w signal intensity, the ADC map, parameters a0 and r of the PUN model applied to DCE imaging [20,24,25]. The 3D color-coded maps were transparently overlaid on the T2w images and readers were asked to classify as positive all CAD marks they considered suspicious for PCa (Figure 1). No other information except the CAD probability map and the T2w imaging were available to the three radiologists during the CAD-assisted reading session. Transparency could be modified using dedicated software (MIPAV v8.0.2, http://mipav.cit.nih.gov, accessed on 27 May 2021). Readers were not allowed to include tumors that were not detected by the CAD system or to discard any suspicious finding based on T2w signal intensities other than findings outside the prostate gland or image artefacts. However, readers could discard findings on the basis of the CAD likelihood map, according to the shape or likelihood value of the candidate region given by the CAD system.

After at least six weeks, readers were asked to report all cases again in a different random order and without the support of the CAD system. During the unassisted interpretation, radiologists reported examinations on the same workstation (Advantage Workstation 4.6, GE Healthcare, Milwaukee, IL, USA) they used in clinical practice, reviewing mpMRI with their favorite reading protocol.

In both assisted and unassisted reading sessions, residents recorded for each suspicious lesion its: (a) location, (b) largest diameter, (c) 5-point confidence score representing the subjective self-confidence that each finding was a tumor (1, absolutely not sure; 5, absolutely sure) and, only in the unassisted reading, the PIRADS score. Interpretation times were also recorded for each reading modality.

### 2.5. Statistical Analysis

The primary aim of this study was to compare per-patient sensitivity, specificity, positive predictive value (PPV) and negative predictive value (NPV) of the CAD-assisted reading with that of the unassisted reporting. In the CAD-assisted mode we considered a positive patient when readers marked as PCa at least one CAD prompt, and negative otherwise. Similarly, in the unassisted reading, a patient was defined positive if at least one lesion scored PIRADS ≥ 3 was found by the radiologists, and negative otherwise. Positive cases on the CAD-assisted/unassisted reading that were confirmed as PCa on pathology were considered true positives, while cases classified negative on mpMRI, but confirmed as PCa on pathology were considered false negatives. Similarly, cases with no findings on mpMRI and no PCa detected on either biopsy or during the follow-up were considered true negative and men with mpMRI reported positive, but not confirmed on either pathology or follow-up were considered false positives.

Per-patient sensitivity was stratified according to Gleason score (GS) in clinically insignificant PCa (GS ≤ 6) and clinically significant ones (GS > 6) and lesion largest diameter (<10 mm vs. ≥10 mm) of the index lesion. Differences in sensitivity, specificity, PPV and NPV between unassisted and CAD-assisted reading were compared using the McNemar test. CAD standalone sensitivities were compared to the average sensitivities across all readers using the chi-squared test.

Secondary analyses focused on the evaluation of the per-lesion sensitivity of the CAD-assisted/unassisted reading. In this case, lesions were considered correctly classified, i.e., true positive, if a finding detected by the radiologists matched the exact location of the pathology-confirmed PCa. The same methods described in the per-patient analysis were applied.

Interpretation times of the two reading sessions were compared using the Mann–Whitney test. Area under receiver operating characteristics (ROC) curves were computed for each reading modality, using the confidence scores given by each reader and the pathology results as the classification variable. Confidence scores were also compared between the two reading sessions using the Mann–Whitney test, while inter-observer agreement between reviewers was evaluated using Fleiss Kappa statistics. All analyses were performed using MedCalc version 15.6.1, except for the Fleiss Kappa statistics which was computed on StatsToDo^©^ (https://www.statstodo.com/CohenFleissKappa_Pgm.php, accessed 27 May 2021). Statistically significance was set at *p* ≤ 0.05.

### 2.6. Power Calculation

Power calculation was based on identifying a change in sensitivity from the CAD-assisted reading to the CAD-unassisted one, since it represents a critical parameter to assess usefulness of a CAD system in detecting PCa. From a previous study in which expert readers reviewed prostate mpMRI with and without a CAD system [20], sensitivity was reported equal to 88% and 81% in the assisted and unassisted modality, respectively, with correlation between paired observation equal to 74%. Sample size was computed with the Mc-Nemar test, by specifying marginal proportions [26], with a power of 80% and a two sided level of significance of 5% and adjusting the sample size for continuity. Therefore, the study would require a sample size of 134 patients to declare that sensitivities obtained on paired proportions are significantly different for all readers combined (approximately 45 cases with positive results for each of the three readers). Then, we considered an equal number of consecutive cases with no finding of PCa, achieving a 1:1 ratio between positive and negative men [27]. Readers were not aware of disease prevalence in this cohort.

## 3. Results

The final dataset comprised 90 patients including 45 men with at least one biopsy-confirmed PCa, for a total of 51 lesions, and 45 patients with no clinical suspicion of PCa for at least 2 years of follow-up. PCa findings were confirmed either by radical prostatectomy (*n* = 34) or prostate biopsy (*n* = 17, of whom *n* = 11 with in-bore biopsy and *n* = 6 with TRUS-guided biopsy). Subjects’ demographics, clinical and pathological findings are reported in Table 1.

### 3.1. Per-Patient Analysis

Per patient specificity and sensitivity in both unassisted and CAD-assisted reading, stratified according to Gleason score and index lesion size is summarized in Table 2. CAD standalone sensitivity, stratified according to Gleason score and index lesion size, was also reported in Table 2. Across all readers, overall sensitivity was statistically not different between the unassisted and CAD-assisted reading sessions (67.4% vs. 70.4%, *p*-value = 0.298). However, sensitivity significantly increased in the CAD-assisted mode for patients with at least one clinically significant lesion (GS > 6) (68.7% vs. 78.1%, *p* = 0.018). When lesions were stratified according to their largest diameters, we did not observe any statistically significant difference between the two reading modalities.

The CAD system standalone correctly detected 43/45 patients (sensitivity = 95.6%, *p* < 0.001 compared to average sensitivity) and over performed results across all readers for each sub-analysis.

Overall specificity was not statistically different between unassisted and assisted sessions (94.8% vs. 89.6, *p* = 0.072) while NPV for unassisted and CAD-assisted reading was 74.4% (95%CI: 69.5–78.8%) and 75.2% (95%CI: 69.9–79.8%), respectively, and PPV 92.9% (95%CI: 86.2–96.4%) and 87.2% (95%CI: 80.3–91.9%), respectively (see also Appendix A).

When reporting with the CAD system, reader 3 obtained a slightly higher sensitivity with respect to reader 1 and 2, considering all lesions (*p* = 0.058). No difference was found in the AUCs representing readers’ confidence for the three readers with and without the CAD system (Figure 2). Again, reader 3 showed a significantly higher AUC than reader 1 and 2 when reporting with the CAD system (*p*-values = 0.029 and 0.017, respectively). 

### 3.2. Per-Lesion Analysis

Per-lesion sensitivity for both unassisted and CAD-assisted reading, stratified by Gleason score and lesion size is summarized in Table 3. Per-lesion CAD standalone sensitivity is also reported in Table 3. Overall, per-lesion sensitivity for both unassisted and assisted reading was 61.4% (94/153). There were no differences between the two reading paradigms when lesions were stratified either according to their GS or lesion size. No differences across readers, with and without CAD, were observed. The CAD system standalone correctly detected 45/51 lesions (sensitivity = 84.6%, *p* = 0.002 compared to average sensitivity) and over performed results across all readers for each sub-analysis. Examples of true positives of the CAD system that were discarded in the CAD-assisted reading either by 2 out of three (Figure 3) or all readers (Figure 4 and Figure 5) are reported. All tumors were also missed in the unassisted reading modality by either all readers (Figure 3 and Figure 4) or 2 out of three readers (Figure 5).

### 3.3. Reading Time and Inter-Reader Agreement

Reading times are summarized in Table 4. Overall, the median reading time of the unassisted and CAD-assisted mode was 170 s (1st–3rd quartile, 101–270 s) and 66 s (1st–3rd quartile, 33–108 s), respectively. Reading times for the CAD-assisted mode were statistically lower than those obtained during the unassisted reading for all readers (*p* < 0.001). Reading times for reporting patients with biopsy-confirmed PCa were statistically higher than those for negative patients for both CAD-unassisted and -assisted reading (*p* < 0.007), except for reader 3, whose reading times in the CAD-assisted modality were not statistically different (*p* = 0.051). 

Table 5 reports inter-observer agreement for both per-patient and per-lesion analysis. In the CAD-assisted mode, a non-significant (*p* = 0.42) increase was observed in the per-patient analysis, while a marginal increase (*p* = 0.06) was found in the per-lesion analysis, compared to the CAD-unassisted reading.

## 4. Discussion

In this study, a statistically significant improvement of per-patient sensitivity in the detection of clinically significant PCa, i.e., with a GS > 6, was observed across all non-experienced readers when reporting mpMRI with CAD, compared to non-assisted reading (*p* = 0.018). Conversely, specificity was not affected (*p* = 0.072). Furthermore, using the CAD system, average reading time was significantly lower (66 s versus 170 s; *p* < 0.001). Interestingly, CAD stand-alone sensitivity was significantly higher than the combined sensitivity of the three non-experienced readers when reporting assisted by the CAD system, meaning that a significant number of true positive findings of the CAD system were erroneously discarded. The latter finding differs from the results of a previous work where experienced readers using a CAD-assisted reading yielded a comparable sensitivity to that of the CAD stand-alone [20]. The cited study also demonstrated that the CAD system significantly improved experienced reader sensitivity in detecting both PCa greater than 10 mm and with a GS > 6. 

By examining CAD true positive cases that were rejected by inexperienced radiologists, we found two possible reasons for the misses. First, we observed that small lesions that were poorly represented by the CAD color coded maps, because they were either small or with irregular shape, were usually discarded by non-experienced readers (Figure 4). Second, lesions overlying a very inhomogeneous background, due to either prostatitis or fibrosis or with a PIRADS 3 score, were also frequently rejected (Figure 5). Of note, on a per-lesion basis, across all readers we observed a non-significant inferiority of sensitivity in CAD-assisted reporting compared to unassisted reading. Our impression is that, again, small additional lesions in patients with multiple PCas could have been discarded by readers on the color-coded CAD maps as they were inconspicuous, while they were correctly identified on mpMRI. 

As previously mentioned for experienced readers [20], different reading behaviors were also observed with inexperienced readers. On a per-patient basis, reader 1 had the lowest sensitivity gain when reporting with the CAD system, while retaining a 100% specificity. Conversely, reader 2 and 3 showed a larger sensitivity increase when assisted by the CAD system but had different results in terms of specificity, i.e., reader 2 performed better without CAD, while only a minimal decrease in specificity was noted for reader 3. Finally, when reporting with the CAD, reader 3 obtained a significantly higher AUC than that of the other 2 readers (*p*-values = 0.029 and 0.017, respectively). While reader 3 definitively showed good performances both sensitivity and specificity-wise, in our opinion the two other readers differed essentially for their trustworthiness. Indeed, while reader 2 was less prone to accept CAD prompts and reported a finding as positive only when he was overly confident, reader 1 was by far much more trustful of CAD prompts which allowed a moderate increase of his sensitivity performances at the unfortunate cost of a significant increase of false positives. Besides individual differences, it must be noted that all three readers rejected a significant number of true positive CAD findings, a far superior number to that of experienced radiologists [20], suggesting that human-CAD interaction was not fully understood and that therefore the potential of the aid system was not being fully used [19]. Other than for their inexperience, the low performances of readers 1 and 3 could also be since they were not accustomed to interpreting mpMRI of the prostate using only color-coded maps provided by the CAD system, superimposed on the T2w imaging. Indeed, prior to entering the study, readers were trained with the CAD system only on 10 cases and were therefore not fully accustomed to the system. While similar human-CAD interactions have been observed in other trials also in different clinical contexts [15,28], Hambrock et al. [12] found that the performance of less-experienced observers was comparable to that of the experienced ones. However, contrary to our study, in the latter study lesions were pre-annotated and readers were aware that all patients had biopsy proven PCa. 

As in our previous trial on experienced readers, here we demonstrate that CAD-assisted reporting significantly reduces reading time [20]. However, results show that there is some evidence that exceedingly short times might affect negatively non-experienced readers’ performances. Indeed, the reader that took more time to report (reader 3) obtained the highest performances.

This study has some limitations. First, results are not generalizable to datasets acquired on different scanners since this was a single center study. Second, readers were asked to either confirm or discard CAD marks; therefore, lesions missed by the CAD system were excluded from the evaluation. Indeed, in this study there were six lesions (five of which smaller than 8 mm) undetected by the CAD system, leading to a maximum reachable per-lesion sensitivity of 84.6%. Among them, four lesions were also missed in the unassisted reading by all readers, while two were detected by only one reader (reader 1 and 3, respectively). Third, the number of readers was small compared to other multi-reader studies. Nevertheless, this work allowed us to recognize the different behaviors of non-experienced reader, providing deeper insight on the human interaction with a CAD system. Fourth, subjects scored PIRADS 1–2 did not undergo biopsy and follow-up was considered to exclude PCa presence. However, we believe that a minimum of 2 years of follow-up with no evidence of PCa is a reasonable time to exclude cancer presence. Finally, we did not correlate mpMRI findings to the molecular changes of these tumors; therefore, results were not stratified based on genetics biomarkers.

## 5. Conclusions

In conclusion, in this study we show that the use of CAD significantly increases the per-patient sensitivity of inexperienced readers in the detection of clinically significant PCa (GS > 6) on mpMRI. Furthermore, in this context CAD does not negatively affect specificity and significantly reduces overall reporting time. 

Several questions will have to be addressed before CAD systems may be reliably used in clinical practice. First, it will be necessary to assess which reading paradigm is more fit in this specific clinical scenario. Indeed, Barinov et al. [29] showed sizeable performance variations of the different reading modalities. Moreover, when evaluating the different CAD paradigms, it will be also necessary to assess the impact of CAD on reporting times [30]. Second, future research should verify if CAD performances remain consistent on datasets originating from different scanners and protocols. Finally, CAD system evaluation will have to be reassessed once a dedicated user-interface and workstation have been developed.

If the results of this study will be validated on a larger dataset, opening the way to use CAD systems in clinical practice, a strong benefit in terms of lower cost/effectiveness, and better detection rate might be reached. Consequently, MRI can be proposed as a screening methodology, both improving patients’ quality of life and strongly decreasing costs and waiting times for the national healthcare system. 

## Figures and Tables

**Figure 1 diagnostics-11-00973-f001:**
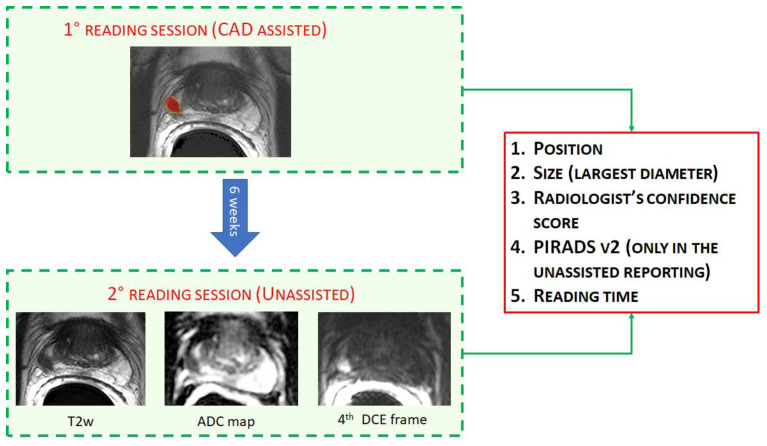
Example of the study workflow. First, cases were reported CAD-assisted, i.e., only the color-coded map overlaid to the T2w image is shown. After 6 weeks, all cases were reported unassisted.

**Figure 2 diagnostics-11-00973-f002:**
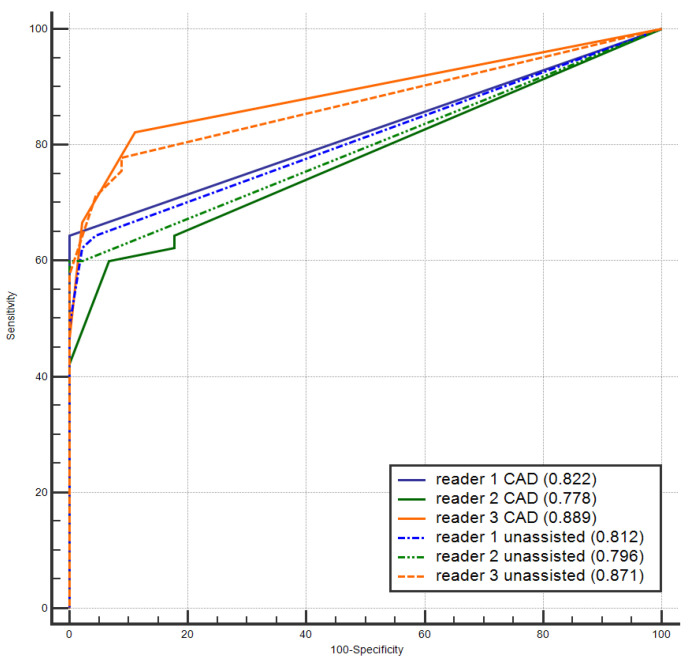
Area under the receiver operating characteristics curve for the three readers for the unassisted and CAD-assisted readings.

**Figure 3 diagnostics-11-00973-f003:**
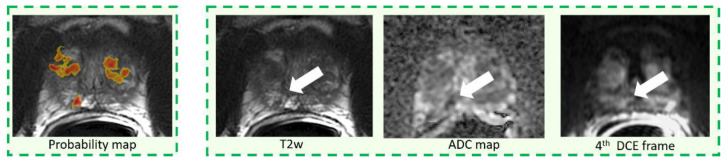
Example of a pGS = 3 + 3 tumor, PIRADS = 3, having largest diameter of 6 mm.

**Figure 4 diagnostics-11-00973-f004:**
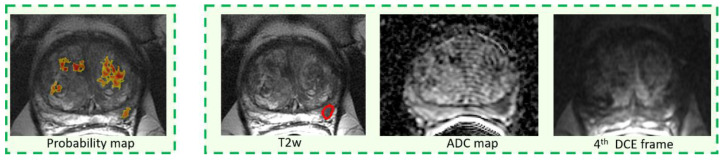
Example of a pGS = 3 + 3 tumor, having largest diameter of 8 mm, and PIRADS = 3.

**Figure 5 diagnostics-11-00973-f005:**
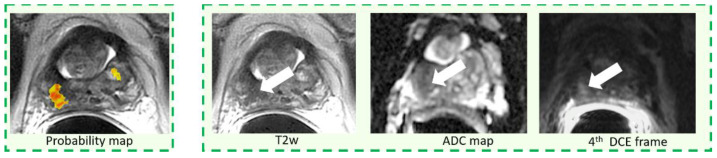
Example of a pGS = 3 + 3 tumor, having largest diameter of 6 mm, and PIRADS = 4.

**Table 1 diagnostics-11-00973-t001:** Patients’ demographics, imaging and pathology findings. Results are presented as either counts or median and interquartile range in parentheses.

	Total	Positive	Negative	*p*-Value
Demographic				
Number of patients, *n* (%)	90	45	45	
Age, y (IQR)	66.7 (63.3–74.7)	68.5 (65.0–75.2)	65.7 (62.0–72.0)	0.059
PSA, ng/mL (IQR)	7.3 (6.0–10.9)	6.9 (5.9–11.9)	7.6 (6.4–10.7)	0.422
Prostate volume, mL (IQR)	52.2 (36.5–80.2)	40.9 (29.6–55.8)	70.2 (48.2–91.0)	<0.001
PSAD, ng/mL/mL (IQR)	0.14 (0.11–0.24)	0.18 (0.13–0.30)	0.13 (0.09–0.17)	<0.001
Imaging				
Longest lesion diameter, mm (IQR)	-	12 (7.8–18)	-	
PI-RADS v2 assessment, *n* (%)				
1	24 (27%)	-	24 (53%)	
2	21 (23%)	-	21 (47%)	
3	4 (5%)	4 (9%)	-	
4	20 (22%)	20 (44%)	-	
5	21 (23%)	21 (47%)	-	
Gleason Score				
3 + 3	13 (29%)	13	-	
3 + 4	16 (36%)	16	-	
4 + 3	10 (22%)	10	-	
4 + 4	3 (7%)	3	-	
4 + 5	1 (2%)	1	-	
5 + 4	1 (2%)	1	-	
5 + 5	1 (2%)	1	-	

**Table 2 diagnostics-11-00973-t002:** Per patient specificity and sensitivity in both unassisted and CAD-assisted reading, expressed as number of percentage and patient/total number of patients in parentheses and corresponding 95% confidence intervals (CIs) in brackets. *p*-values in bold are statistically significant.

	Unassisted Reading (%)	Assisted Reading (%)	*p* Value
Sensitivity			
Reader 1	64.4 (29/45) [48.8–78.1]	64.4 (29/45) [48.8–78.1]	0.500
Reader 2	60.0 (27/45) [44.3–74.3]	64.4 (29/45) [48.8–78.1]	0.387
Reader 3	77.8 (35/45) [62.9–88.8]	82.2 (37/45) [67.9–92.0]	0.344
Average	67.4 (91/135) [58.8–75.2]	70.4 (95/135) [61.9–77.9]	0.298
CAD standalone	-	95.6 (43/45) [84.8–99.5]	**<0.001**
Sensitivity for GS = 6			
Reader 1	61.5 (8/13) [31.6–86.1]	46.2 (6/13) [19.2–74.9]	0.363
Reader 2	53.8 (7/13) [25.1–80.8]	38.5 (5/13) [13.9–68.4]	0.344
Reader 3	76.9 (10/13) [46.2–95.0]	69.2 (9/13) [38.6–90.9]	0.500
Average	64.1 (25/39) [47.2–78.8]	51.3 (20/39) [34.8–67.6]	0.166
CAD standalone	-	92.3 (12/13) [64.0–99.8]	**0.005**
Sensitivity for GS > 6			
Reader 1	65.6 (21/32) [46.8–81.4]	71.9 (23/32) [53.2–86.2]	0.344
Reader 2	62.5 (20/32) [43.7–78.9]	75.0 (24/32) [56.6–88.5]	0.109
Reader 3	78.1 (25/32) [60.0–90.7]	87.5 (28/32) [71.0–96.5]	0.125
Average	68.7 (66/96) [58.5–77.8]	78.1 (75/96) [68.5–85.9]	**0.018**
CAD standalone	-	95.6 (31/32) [78.1–99.9]	**0.012**
Sensitivity for max diameter 4–9 mm			
Reader 1	41.1 (7/17) [18.4–67.1]	52.9 (9/17) [27.8–77.0]	0.344
Reader 2	41.1 (7/17) [18.4–67.1]	47.1 (8/17) [23.0–72.2]	0.500
Reader 3	64.7 (11/17) [38.3–85.8]	76.5 (13/17) [50.1–93.2]	0.313
Average	49.0 (25/51) [34.7–63.4]	58.8 (30/51) [44.2–72.4]	0.151
CAD standalone	-	94.1 (16/17) [71.3–99.8]	**0.004**
Sensitivity for max diameter ≥ 10 mm			
Reader 1	78.6 (22/28) [59.0–91.7]	71.4 (20/28) [51.3–86.8]	0.363
Reader 2	71.4 (20/28) [51.3–86.8]	75.0 (21/28) [55.1–89.3]	0.500
Reader 3	85.7 (24/28) [67.3–96.0]	85.7 (24/28) [67.3–96.0]	0.500
Average	78.6 (66/84) [68.3–86.8]	77.4 (65/84) [66.9–85.8]	0.500
CAD standalone	-	96.4 (27/28) [81.6–99.9]	**0.012**
Specificity			
Reader 1	95.6 (43/45) [84.9–99.5]	100 (45/45) [92.1–100.0]	0.250
Reader 2	97.8 (44/45) [88.2–99.9]	80.0 (36/45) [65.4–90.4]	**0.004**
Reader 3	91.1 (41/45) [78.8–97.5]	88.9 (40/45) [75.9–96.3]	0.500
Average	94.8 (128/135) [89.6–97.9]	89.6 (121/135) [83.2–94.2]	0.072

**Table 3 diagnostics-11-00973-t003:** Per-lesion sensitivity expressed as percentage and number of patient/total number of patients in parentheses and corresponding 95% confidence intervals (CIs) in brackets. *p*-values in bold are statistically significant.

	Unassisted Reading	Assisted Reading	*p* Value
Sensitivity			
Reader 1	60.8 (31/51) [46.1–74.2]	54.9 (28/51) [40.3–68.9]	0.304
Reader 2	52.9 (27/51) [38.5–67.1]	56.9 (29/51) [42.2–70.6]	0.387
Reader 3	70.6 (36/51) [56.2–82.5]	72.6 (37/51) [58.3–84.1]	0.500
Average	61.4 (94/153) [53.2–69.2]	61.4 (94/153) [53.2–69.2]	0.500
CAD standalone	-	84.6 (45/51) [76.1–95.6]	**0.001**
Sensitivity for GS = 6			
Reader 1	57.1 (8/14) [28.9–82.3]	42.9 (6/14) [17.7–71.1]	0.363
Reader 2	50.0 (7/14) [23.0–77.0]	35.7 (5/14) [12.8–64.9]	0.344
Reader 3	71.4 (10/14) [41.9–91.6]	71.4 (10/14) [41.9–91.6]	0.500
Average	59.6 (25/42) [43.3–74.4]	50.0 (21/42) [34.2–65.8]	0.240
CAD standalone	-	92.9 (13/14) [66.1–99.8]	**0.002**
Sensitivity for GS > 6			
Reader 1	62.2 (23/37) [44.8–77.5]	59.5 (22/37) [42.1–75.2]	0.500
Reader 2	54.1 (20/37) [36.9–70.5]	64.9 (24/37) [47.5–79.8]	0.109
Reader 3	70.3 (26/37) [53.0–84.1]	73.0 (27/37) [55.9–86.2]	0.500
Average	62.2 (69/111) [52.4–71.2]	65.8 (73/111) [56.2–74.5]	0.240
CAD standalone	-	86.5 (32/37) [71.2–95.5]	**0.008**
Sensitivity for max diameter 4–9 mm			
Reader 1	39.1 (9/23) [19.7–61.5]	39.1 (9/23) [19.7–61.5]	0.500
Reader 2	34.8 (8/23) [16.4–57.3]	39.1 (9/23) [19.7–61.5]	0.500
Reader 3	56.6 (13/23) [34.5–76.8]	60.9 (14/23) [38.5–80.3]	0.500
Average	43.5 (30/69) [31.6–56.0]	46.4 (32/69) [34.3–58.8]	0.407
CAD standalone	-	78.2 (18/23) [56.3–92.5]	**0.004**
Sensitivity for max diameter ≥ 10 mm			
Reader 1	78.6 (22/28) [59.0–91.7]	67.9 (19/28) [47.6–84.1]	0.254
Reader 2	67.9 (19/28) [47.6–84.1]	71.4 (20/28) [51.3–86.8]	0.500
Reader 3	82.1 (23/28) [63.1–93.9]	82.1 (23/28) [63.1–93.9]	0.500
Average	76.2 (64/84) [65.6–84.8]	73.8 (62/84) [63.1–82.8]	0.407
CAD standalone	-	96.4 (27/28) [81.6–99.9]	**0.005**

**Table 4 diagnostics-11-00973-t004:** Interpretation time for CAD-unassisted and -assisted reading expressed as median values, with interquartile range in parentheses.

	Unassisted Reading		*p* Value	Assisted Reading		*p* Value
Reader 1	146 (91–199)			35 (25–77)		<0.001
Reader 2	120 (84–210)			70 (37–99)		<0.001
Reader 3	255 (180–403)			100 (56–180)		<0.001
Average	170 (101–270)			66 (33–108)		<0.001
	Biopsy +	Biopsy −		Biopsy +	Biopsy −	
Reader 1	187 (135–281)	122 (90–158)	<0.001	70 (47–99)	25 (20–35)	<0.001
Reader 2	180 (117–233)	90 (60–129)	<0.001	80 (49–106)	50 (30–90)	0.007
Reader 3	331 (224–473)	200 (127–291)	<0.001	120 (70–205)	98 (40–170)	0.051
Average	210 (143–325)	125 (90–200)	<0.001	90 (50–120)	40 (25–90)	<0.001

**Table 5 diagnostics-11-00973-t005:** Inter-observer agreement between reviewers evaluated using Fleiss Kappa statistics. 95% confidence intervals (CIs) are reported in parentheses.

	Per-Patient Analysis		Per-Lesion Analysis	
CAD-Assisted	Reader 1	Reader 3	Reader 1	Reader 3
Reader 2	0.647 (0.488–0.806)	0.641 (0.482–0.799)	0.722 (0.531–0.913)	0.631 (0.427–0.834)
Reader 3	0.704 (0.561–0.846)	-	0.582 (0.364–0.800)	-
Overall	0.662 (0.542–0.781)		0.641 (0.483–0.780)	
Unassisted	Reader 1	Reader 3	Reader 1	Reader 3
Reader 2	0.746 (0.598–0.893)	0.602 (0.438–0.766)	0.603 (0.385–0.821)	0.441 (0.188–0.693)
Reader 3	0.605 (0.440–0.770)	-	0.397 (0.159–0.635)	-
Overall	0.646 (0.527–0.765)		0.476 (0.317–0.634)	

## Data Availability

The data presented in this study are available on request from the corresponding author. The data are not publicly available due to privacy.

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
