# Peer review of "Computer-Aided Diagnosis Improves the Detection of Clinically Significant Prostate Cancer on Multiparametric-MRI: A Multi-Observer Performance Study Involving Inexperienced Readers"

_diagnostics, 2021, doi:10.3390/diagnostics11060973_

Round 1

Reviewer 1 Report

Authors have shown the use of the Computer Aided Diagnosis system which significantly increases the per-patient sensitivity of inexperienced readers in the detection of clinically significant prostate cancer, without negatively affecting specificity, while significantly reducing overall reporting time.

What is the method used for power calculation?

What will be the cost of this system and clinical relevance needs to be addressed in the discussion?

Can this method be used for other cancers?

Is there any molecular changes that can be correlated?

Author Response

Authors have shown the use of the Computer Aided Diagnosis system which significantly increases the per-patient sensitivity of inexperienced readers in the detection of clinically significant prostate cancer, without negatively affecting specificity, while significantly reducing overall reporting time.

 We thank the reviewer for reading our work and providing interesting issues that we addressed.

Q: What is the method used for power calculation?

A: We thank the reviewer for this valuable comment. Previously, we only reported the reference of the method, however, he/she is right that to better understand the text it is important to also describe the method in the materials and methods section. We add the requested information.

Q: What will be the cost of this system and clinical relevance needs to be addressed in the discussion?

A: We thank the reviewer for this important question. The cost of this system is not very high since it only requires a server to run the algorithms. However, we did not assess this issue in the discussion since it would require completely different study. However, we added in the conclusions section a paragraph related to the clinical relevance of our method.

Q: Can this method be used for other cancers?

A: The methods described in this paper were specifically developed to diagnose prostate cancer. Similar CAD systems were previously developed (e.g. CAD colon and CAD lung, as we better specified in the revised introduction) and also our CAD can be applied to other cancers using a similar pipeline for training and validation. However, the aim of this paper was not to develop a CAD system but to demonstrate that using a single image (i.e., the probability map) instead of the whole multi-parametric sequence can both improve sensitivity and reduce reading time. Therefore, it would be possible that the same strategy can be used for other cancers.

Q: Is there any molecular changes that can be correlated?

We thank the reviewer because this is an important issue that could have been very interesting to evaluate. However, unfortunately these patients did not undergo any genetics/molecular analysis. We discussed this point in the discussion section.

Reviewer 2 Report

In this paper, three resident radiologists were asked to review multiparametric-MRI of patients with and without PCa, both unassisted and assisted by a CAD system. In both reading sessions, residents recorded all positive cases, and sensitivity, specificity, negative and positive predictive values were computed and compared.

However, this paper in the current format is not acceptable and needs serious major revision.

  1. The topic of your paper is too long. Please summarize your topic.
  2. The introduction is too low. Please explain more about your research work and research gap.
  3. Divide introduction into introduction and background.
  4. You only have 20 references which are too low. Please add at least 10 references in the background.
  5. Most of the references are related to many years ago. Please consider references related to recent years(2019-2020-2021).
  6. The topic for tables 2 and 3 is pretty weird and unusual and it is a paragraph. Please consider your explanation in the context and provide just one line for the second and third tables.
  7. The topic for all figures is pretty weird and unusual and it is a paragraph. Please consider your explanation in the context and provide just one line for the second and third tables.
  8. I do not know why after figure 2, you consider figure. Please do according to ordering numbers.
  9. Your discussion is too long and your conclusion is too low. Please add more content to your conclusion and consider future works as well.

Author Response

In this paper, three resident radiologists were asked to review multiparametric-MRI of patients with and without PCa, both unassisted and assisted by a CAD system. In both reading sessions, residents recorded all positive cases, and sensitivity, specificity, negative and positive predictive values were computed and compared.

However, this paper in the current format is not acceptable and needs serious major revision.

We thank the reviewer for the valuable comments that he/she provided. We changed the text accordingly, and we will provide a point-by-point response to all questions.

1. The topic of your paper is too long. Please summarize your topic.

A: We thank the reviewer, but we are not quite sure to have understood the comment. In particular, we are confused about the term “topic”. If the reviewer was referring to the title, we shortened it.

2.  The introduction is too low. Please explain more about your research work and research gap.

A: We thank the reviewer for this valuable comment. We better contextualize the field (first paragraph) and we extended the research field and gap, as suggested. However, we decided to keep the introduction short and to keep the description of our work in the material section, as suggested by the guidelines and as it has been done by to other similar works (e.g., Greer, M.D.; Lay, N.; Shih, J.H.; Barrett, T.; Bittencourt, L.K.; Borofsky, S.; Kabakus, I.; Law, Y.M.; Marko, J.; Shebel, H.; et al. Computer-aided diagnosis prior to conventional interpretation of prostate mpMRI: an international multi-reader study. Eur. Radiol. 2018, 28, 4407–4417, doi:10.1007/s00330-018-5374-6.)

3. Divide introduction into introduction and background.

A: We organized the introduction as the reviewer suggested, describing the background and the current diagnostic workup for prostate cancer patients. Then, we highlighted the limitations which we wanted to address in our work.

4. You only have 20 references which are too low. Please add at least 10 references in the background.

A: We thank the reviewer, we added 10 related references, especially in the introduction and in the conclusion section.

5. Most of the references are related to many years ago. Please consider references related to recent years (2019-2020-2021).

A: We added 7 references from 2019 to 2021

6. The topic for tables 2 and 3 is pretty weird and unusual and it is a paragraph. Please consider your explanation in the context and provide just one line for the second and third tables.

A: The reviewer is right. We shortened the topic of both tables and revised the text accordingly. We also add part of the description in the statistical analysis section, since it belonged to that section.

7. The topic for all figures is pretty weird and unusual and it is a paragraph. Please consider your explanation in the context and provide just one line for the second and third tables.

A: The reviewer is right. We modified the title and the text of the figures accordingly.

8. I do not know why after figure 2, you consider figure. Please do according to ordering numbers.

A: We are not quite sure we correctly understand this comment. However, we checked that the order of the figures was correct.

9. Your discussion is too long and your conclusion is too low. Please add more content to your conclusion and consider future works as well.

A: We thank the reviewer, however we partially disagree with him/her. We added some content on the conclusion section (i.e., clinical relevance and future works), but we decided to keep the discussion as they were since we commented the results obtained and contextualized them, and we assessed the limitations of our work. All these topics are requested by the guidelines as following reported:

“Authors should discuss the results and how they can be interpreted in perspective of previous studies and of the working hypotheses. The findings and their implications should be discussed in the broadest context possible and limitations of the work highlighted.”

Round 2

Reviewer 1 Report

Manuscript can be accepted in its current form.

Reviewer 2 Report

This version is available for the publication